# Presence and Natural Treatment of Organic Micropollutants and their Risks after 100 Years of Incidental Water Reuse in Agricultural Irrigation

**Alma C. Chávez-Mejía \*, Inés Navarro-González, Rafael Magaña-López, Dafne Uscanga-Roldán, Paloma I. Zaragoza-Sánchez and Blanca Elena Jiménez-Cisneros**

Instituto de Ingeniería, UNAM, Circuito Escolar s/n, Delegación Coyoacán,
Ciudad de Mexico D.F. CP 04510, Mexico; INavarroG@iingen.unam.mx (I.N.-G.);
RMaganaL@iingen.unam.mx (R.M.-L.); DUscangaR@iingen.unam.mx (D.U.-R.);
PZaragozaS@iingen.unam.mx (P.I.Z.-S.); BJimenezC@iingen.unam.mx (B.E.J.-C.)
**\*** Correspondence: AChavezM@iingen.unam.mx; Tel.: +52-5556-2336-00

**Abstract:** The aim of the research was to show the presence of micropollutants contained in the wastewater of Mexico City within the distribution canals of the Mezquital Valley (MV), as well as their retention in agricultural soil and aquifers. This system constitutes the world's oldest and largest example of the use of untreated wastewater for agricultural irrigation. The artificial recharge associated with the MV aquifers, with groundwater extracted for human consumption showing its importance as a water resource for Mexico City. The results of sampling show the presence of 18 compounds, with 10 of these considered as endocrine disruptor compounds (EDCs). The concentration of these pollutants ranged from 2 ng/L for 17 β-estradiol to 99 ng/L for DEHP, with these values decreasing throughout the course of the canals due to the wastewater dilution factor, their retention in agricultural soil, and their accumulation in the local aquifer. The main mechanisms involved in natural attenuation are adsorption, filtration, and biodegradation. Drinking water equivalent levels (DWELs) were estimated for 11 compounds with regard to acceptable daily intakes (ADIs), by assuming local exposure parameters for a rural Mexican population. These were compared with the maximum groundwater concentrations (Cgw) to screen the potential risks. The very low ratios of Cgw to DWELs indicate no appreciable human health risk from the presence of trace concentrations of these compounds in the source of drinking water in the MV. Despite this, far from being exceeded after more than 100 years of irrigation with residual water, the natural soil attenuation seems to remain stable.

**Keywords:** residual water; organic micropollutants; natural treatment; Mexico City; Mezquital Valley; agricultural irrigation; environmental risk

## 1. Introduction

The worldwide trend toward urbanization is increasing the volume of untreated and treated wastewater. These effluents need to be disposed into the environment and contain a new generation of micropollutants commonly known as emerging pollutants. Wastewater is the most common source of these compounds and this has created several concerns among scientists and policy makers dealing with water use. These include: (a) the reuse of water for human consumption, either intentionally or unintentionally, (b) the reuse of wastewater for irrigation and where aquifers are recharged indirectly through this activity, (c) in soil-aquifer treatment systems, (d) when aquifers are intentionally recharged in order to increase the volume of water sources, and (e) the need to change the concept of wastewater disposal to that of reintegrating used wastewater to the environment.

Irrigation of farmland with wastewater has a long history and many cities around the world, which often lack adequate infrastructure to treat wastewater, use it as a relatively inexpensive management option. The Mezquital Valley system (85000 ha) is the world's oldest and largest project where urban untreated wastewater has been used for agricultural irrigation [1]. At the end of the 19th century (1896), domestic, pluvial, and industrial water was redirected through three distribution canals to prevent flooding in the Mexico Valley: the "Gran Canal" (known as Grand Canal or Grand Channel) (1898), the "Interceptor del Poniente" (Eastern Emmiter) (1989), and the "Emisor Central" (Central Emitter) (1975). They send the content to an area known as "the Tula Valley" (90 km north of the city). A fourth channel is under construction, and is called the "Túnel Emisor del Oriente" (Western Emmiter Tunnel). It will have a capacity of 60 m³/s and will begin operation in 2020. The Tula Valley, which is also known as the "Mezquital Valley," receives a total of 52 m³/s. Additionally, 70% of this is wastewater generated within Mexico City (41 m³/s), while the other 30% remains in two irrigation zones located in the Mexico Valley. Most of the untreated wastewater is used to supply irrigated areas and has undergone a gradual increase related to population growth. In 1926, an irrigated area of 14,000 ha was reported and, by 2010, this had increased to 85,000 ha. This is equivalent to a doubling rate of the irrigation surface every 25 years [2,3]. These changes led to the development of three irrigation districts (ID): "Alfajayucan" (ID-100), "Ajacuba" (ID-112), and "Tula" (ID-03), as shown in Figure 1.

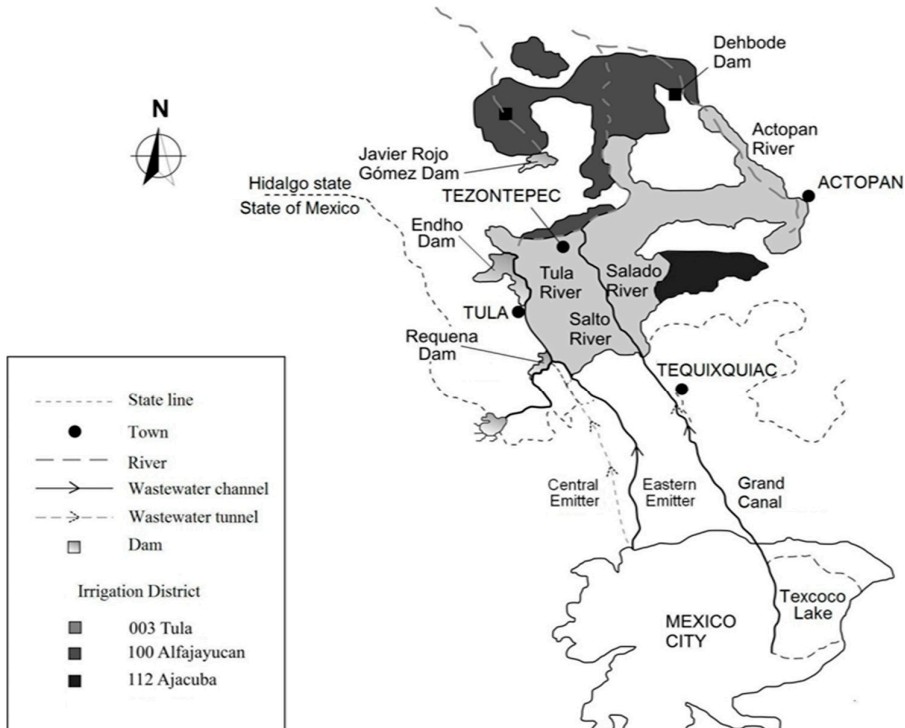

**Figure 1.** Transportation of wastewater generated in the Mexico Valley and sent to the Tula Valley (Source: "Grupo Tratamiento y Reúso del Agua").

It has been estimated that the system's current recharge rate is around 800 Mm³/year (25 m³/s, 2160 MLD), which represents a thirteen-fold increase when compared to the original flowrate. Due to the use of large irrigation rates and the prevailing soil conditions in the valley, remarkably large water volumes began to artificially recharge the system, forming a shallow aquifer. Because of the artificially high recharge rate associated with the local aquifers, groundwater is now extracted for human consumption, and there are plans to use this groundwater as a water resource for Mexico City. Within this zone, there are 283 water sources available to supply the population (112 wells and 22 springs). Of the total extracted volume of water, which is around 232 m³/s, 64% is used by industry, 22% is used by agriculture, and 14% is used for human consumption [1]. New artesian wells and

springs with flows ranging from 40 to 600 L/s appeared from 1964 onwards. A result of the water table rising from a level 50 m below the surface. Currently, 6 m$^3$/s of water extracted from the subsoil is used in the valley, 17% for human consumption, 38% for agriculture, 33% for industry, and 12% for other uses [4], while 21 m$^3$/s still flows to the neighboring basin for reuse in agriculture.

The quality of the water has been evaluated according to the parameters established by the local regulations since sewage began to be conveyed from the Mexico Valley. As a result, only some relevant parameters are assessed, such as basic indicators of organic matter, nutrients, and those related to fecal pollution, with no wastewater standard re-drafting for more than 20 years. As a result, the standard does not consider a variety of currently relevant organic and emerging micropollutants and other regulated substances.

Recent advances in chromatographic separation techniques and mass selective detection have confirmed the presence of organic micropollutants (OMPs) in environmental matrices (surface water, groundwater, soil, sediments, biota, and suspended particles), which allows a range of concentrations to be established for some of these contaminants [5], but not all. Their detection on the global scale is low, since monitoring studies are concentrated in 30 countries [6], and most of these are located in Europe and North America. In this case, the wastewater generated is usually treated to the secondary level (thereby reducing the incidence of these pollutants considerably). Such studies are not common in countries with emerging economies, which results in information gaps concerning concentrations and environmental targets, particularly in places where untreated wastewater is released (such as China and Mexico). This situation can be attributed to analytical limitations, as well as the allocation of economic and human resources for the development of field and laboratory surveys and the implementation of treatment technologies [7–10].

The impact on human health of persistent organic pollutants (POPs) may be highlighted in terms of exposure and the effects of endocrine disruptors. It is known that exposure to endocrine disruptor compounds (EDCs) could play a significant role in the causality of many more endocrine diseases and disorders than previously thought. Examples of these include: female reproductive dysfunction, effects on male reproductive health, adrenal disorders, and the development of immune system issues, thyroid related disorders, neuro-developmental dysfunctions in children, endocrine-related cancers, and metabolic and bone disorders [11–13]. There is currently no widely agreed system to assess the strength of associations between chemical exposure (including EDCs) and adverse health outcomes [14].

The need to develop improved approaches to assess the evidence, along with improved methods of risk assessment, has been widely recognized. The health risks associated with pharmaceuticals and EDCs in water have been assessed mainly by using two approaches: the minimum therapeutic dose (MTD) and the acceptable daily intake (ADI). In addition, the drinking water equivalent levels (DWELs) are sometimes considered. They have been used as reference values to obtain a margin of safety between a given sample and the worst reported or predicted case exposure in drinking water [14]. The MTD is usually lower than concentrations where unacceptable, adverse, or toxic effects are rarely observed. However, the ADI is an amount that can be ingested daily for a long period of time, usually over a lifetime, without a significant health risk. This latter approach will be used in this case study. Consequently, the present research aims to demonstrate the presence, destination, and attenuation mechanisms of a group of OMPs found in water, as well as to assess their potential for human health risks using Mexico City as a case study.

Since the year 2000, interest in evaluating these organic compounds in Mexico has increased greatly, which is highlighted in the studies conducted within References [15–17]. Conversely, few works have evaluated the content of these organic contaminants in wastewater and groundwater in the same area. Those that did measured very few compounds [18]. They tended only to assess the occurrence and fate of semi-volatile organic compounds, pharmaceutically active compounds, and other emerging contaminants in the Tula Valley and focused on their accumulation and dissipation, or

their sorption/desorption from soils [18]. None of them established the human health risks that they represent. The present study intends to do so.

The previously mentioned factors pose a new challenge to environmental scientists. Previous studies have considered the removal, fate, and transportation of what may be termed conventional pollutants, instead of the disparate mixture of many classes of compounds with widely varying properties that have been labeled as emerging compounds. To contribute to the research, it is necessary to review what we know and what we do not know concerning emerging pollutants, especially with regard to their fate and transportation, which will ultimately define risks.

The objective of this work was to determine the content of micropollutants for the different sources of the unique Tula Valley system, and to assess the impact of their use in irrigation with emphasis on human health. This site is of the utmost importance, since unintentional recharge has been carried out for more than a century due to the practice of untreated wastewater irrigation. Several new springs have been generated, including the Cerro Colorado spring that now supplies water to more than 500.000 habitants. This water is treated by chlorine disinfection since this is the only available method of purification (regardless of origin). The presence of EDCs has been reported in the spring water and in the supply wells of the region, despite the clear effect of natural soil treatment on the attenuation of contaminants from the wastewater originally used to irrigate. No previous studies have been carried out in the region to determine the presence of these compounds in the harvested crops. These include "alfalfa" (*Medicago* sp.), corn, wheat, barley, forage oats, tomatoes, and chili, among others. No data is available concerning the productivity, size, and quality of crops or indications that show any appreciable impact of these emerging pollutants on them. No previous work in Mexico has evaluated such a large area of influence. The present research integrates and analyzes the zone through the compilation of 10 years of results from studies of various environmental matrices. In addition, it aims to serve as a precedent for future research in diverse fields such as environmental sciences, sustainable development, economics, and politics. The integrated study serves to determine the relevance of assessing emerging contaminants in wastewater for subsequent uses, as well as to reveal the possible risk for human health of the water management project within the Tula Valley.

## 2. Materials and Methods

### 2.1. Sampling

For the purposes of this study, three zones of the Tula Valley were considered for sampling, based on the properties of the water used. Three different regions of the Tula valley were identified where irrigation was predominantly either carried out with untreated wastewater (sewage from Mexico City) used directly for irrigation soon after its arrival in the valley (zone 1), raw wastewater diluted with rainwater or previously stored in dams (zone 2), and raw wastewater mixed with spring or well water (zone 3).

As a starting point, the content of OMPs present in wastewater from the "Emisor central" (Central Emitter) feeding the main distribution canals of the Tula Valley was analyzed, together with the main sources of supply, including dug wells ("Norias"), springs (low aquifer), and wells (deep aquifer) located on the three zones of influence labeled as Zone 1, 2, and 3 (Figure 2).

The first zone was in the south of the region with use occurring approximately 20 to 24 h after leaving the city. The second was further to the north where the wastewater has traveled further and some (amounts depend on demand from farmers) has passed through a storage/flow regulation dam (the Endho Dam) and has been mixed with stored rainwater (such as from the Requena Dam) prior to irrigation. In the third zone, the same wastewater as used in the second zone has traveled further up the valley (up to 48 h travel time from the city if not stored in the Endho Dam) and is mixed with well water before being used for irrigation.

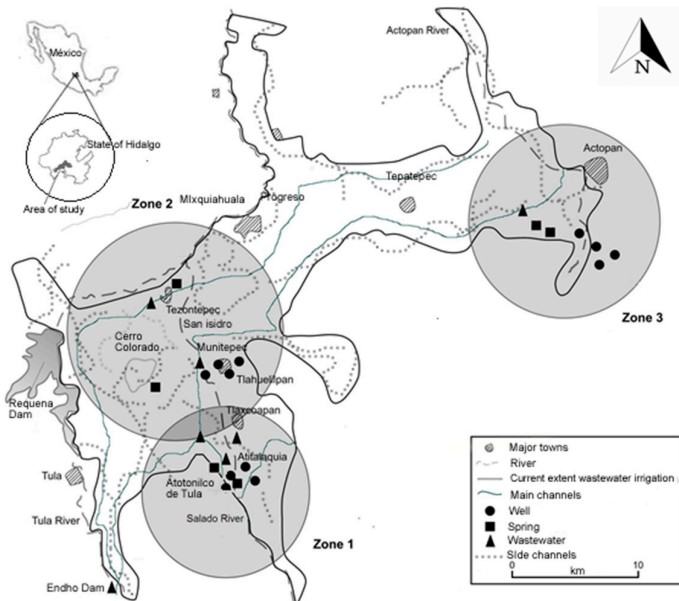

**Figure 2.** Location of sampling points according to the study area. (Source: "Grupo Tratamiento y Reúso del Agua").

Samples were taken from four wells (water extracted by a pump), one dug well (water extracted by hand), and one or two springs. In total, infiltrated water was sampled from 12 wells, four springs, and two dug wells. For each area and for each point, the samples were taken in triplicate over two campaigns with each lasting six weeks at different periods of the year: rainy season (May to October) (2007) and dry season (November to April) (2008). Further details are given in References [16,17,19]. Wastewater was sampled from the Central Emitter (the principal canal that delivers wastewater from the drainage system of Mexico City to the Tula Valley, and named as Ref. ZERO) and distribution canals from the three different zones described previously. The water was sampled directly into clean glass bottles, transferred to the laboratory, and stored at 4 °C overnight before processing the next day. Each sampling point was assigned a tag that identified the study zone, the type of water, and an individual number (Table 1).

**Table 1.** Specific details of the sampling points for the three zones (name, location, and identification tag).

| Sampling Points | Name Local Name of Identification | West | North | * Distance (km) |
|---|---|---|---|---|
| Zone 1. Raw wastewater from Mexico City without any treatment | | | | |
| Well 1 | Tezoquipa | 98°47′31″ | 20°3′30″ | 55.07 |
| Well 2 | Principal la Cantera | 98°48′31″ | 20°3′11″ | 52.37 |
| Well 3 | El Tablón | 98°47′11″ | 20°2′53″ | 54.53 |
| Well 4 | Pozo San Primitivo | 98°47′7″ | 20°7′47″ | 57.13 |
| Spring 1 | Rancho Nápoles (las albercas) | 98°46′40″ | 20°2′22″ | 55.24 |
| Dug well 1 | Fam. González López | 98°47′32″ | 20°3′29″ | 54.17 |
| Central Emmiter | Emisor Central | 99°17′49″ | 19°56′58″ | Ref. ZERO |
| WW Canal 1 | Canal de Riego (Tequizquiac) | 99°15′11″ | 20°4′24″ | 14.46 |
| WW Canal 2 | Canal Salto Tlamaco km 22 | 99°15′11″ | 20°4′2″ | 13.82 |

**Table 1.** *Cont.*

| Sampling Points | Name Local Name of Identification | West | North | * Distance (km) |
|---|---|---|---|---|
| Zone 2. Raw wastewater diluted with rainwater or previously stored in dams | | | | |
| Well 1 | Pozo no. 9 del sistema 2 de CFE | 98°44′60″ | 20°6′40″ | 59.95 |
| Well 2 | Pozo no. 11 del sistema 2 de CFE | 98°45′30″ | 20°8′14″ | 60.06 |
| Well 3 | Pozo Miravalle | 99°13′21″ | 20°8′21″ | 22.40 |
| Well 4 | Pozo 5 de mayo | 99°13′16″ | 20°7′42″ | 21.33 |
| Spring 1 | Cerro Colorado | 98°44′16″ | 20°7′13″ | 61.48 |
| Spring 2 | Manantial Puedhe | 98°43′15″ | 20°11′28″ | 65.94 |
| WW Canal 1 | Canal Principal Requena | 99°14′6″ | 20°6′37″ | 18.95 |
| WW Canal 2 | Canal de Riego | 99°15′57″ | 20°8′44″ | 21.95 |
| Zone 3. Raw wastewater mixed with spring or well water | | | | |
| Well 1 | Bothi Baji | 99°3′34″ | 20°13′53″ | 38.89 |
| Well 2 | San Isidro | 99°2′43″ | 20°16′25″ | 44.50 |
| Well 3 | CIC 126 | 99°3′33″ | 20°12′21″ | 37.74 |
| Well 4 | Pozo CIC 14 | 99°3′46″ | 20°13′7″ | 38.57 |
| Spring 1 | Manantial El Dren | 99°1′44″ | 20°16′25″ | 45.54 |
| Dug well 1 | Noria de Pozo Grande | 98°57′30″ | 20°15′54″ | 49.74 |
| WW Canal 1 | Agua de riego | 98°58′10″ | 20°12′13″ | 37.50 |

WW = Wastewater, * = Approximately distance from the Ref. Zero, Ref. Zero = Effluent of residual water (main point of discharge).

## 2.2. Estimation of OMPs

For the analysis of organic micropollutants, all water samples were stored at 4 °C overnight and extracted within 24 h. The determination was performed in accordance with the method validated by Reference [16]. Briefly, the sample was acidified (pH = 2) with concentrated $H_2SO_4$. The micropollutants were isolated from the water samples using OASIS HLB filter cartridges (200 mg), previously conditioned with $2 \times 5$ mL of acetone and 5 mL of acetic acid 5%. The acidic pharmaceuticals and carbamazepine were recovered with 5.5 mL of a 40:60 solution of acetone: buffer $NaHCO_3$ buffer (pH = 10), whereas the phenolic compounds were eluted using 5 mL of acetone. The silylated derivates of acid pharmaceuticals were obtained using the derivatizer MTBSTFA, whereas the phenol dimethylsilyl esters were obtained using the agent BSTFA. The analytes were analyzed and quantified in an HP 6890 gas chromatograph coupled to a selective mass detector fitted with a 30 m HP5-MS fused silica capillary column (30 m × 0.25 mm, 0.25 μm film thickness), connected to an HP 5397. 2,3-dichlorophenoxyacetic acid (2,3-D), [2H4] 4-n-nonylphenol and [2H16] bisphenol-A were used as internal standards. The analysis quality control was guaranteed using recovery standards of 3,4-dichlorophenoxyacetic (3,4-D) acid, 4-n-nonylphenol, and 10-11 dihydrocarbamazepine. For each batch, blanks were used to remove the analytes added throughout the analytical determination.

## 2.3. Estimation on Human Health Risks

As indicated above, the acceptable daily intake (ADI) approach, which has been adopted by others [20–23], was used in this study. The worst-case scenario of the potential health risk of seven EDCs detected in groundwater (i.e., the source for drinking water) in the Mezquital Valley were estimated by using the maximum concentrations (Cwg, μg/L) recorded (Table 2).

**Table 2.** Concentrations and acceptable daily intakes (ADIs) used for EDC risk analysis.

| Phase | Compound | Class | Effects | Maximum Concentration Cgw | ADI Reported |
|---|---|---|---|---|---|
| | OMP | | | μg/L | μg/kg-day |
| Acids | Gemfibrozil | Antilipidemic | Developmental | $2.40 \times 10^{-4}$ | 31 |
| | Naproxen | NSAID | Reproductive/Developmental | 0.012 | 570 |
| | Diclofenac | NSAID | Developmental | $6.20 \times 10^{-4}$ | 67 |
| | 4-n-nonylphenols * | Surfactant | Developmental | 0.075 | 50 |
| | 17β-estradiol * | EES | Endocrine | $7.00 \times 10^{-5}$ | 0.05 |
| Phenols | Bisphenol-A * | Chemical industry | Developmental | 0.171 | 50 |
| | Butylbenzil-phthalate (BuBeP) * | Phthalate plasticizer | Reproductive/Developmental | 19.55 | 100 |
| | Carbamazepine * | Anticonvulsant | Developmental | 0.175 | 10 |
| | Di-2(ethilhexyl)-phthalate (DEHP) * | Phthalate | Reproductive/Developmental | 2.021 | 12 |
| | Estrone * | EES | Endocrine/Liver | $2.60 \times 10^{-4}$ | 0.013 |

NSAID (non-steroidal anti-inflammatory drug), EES (Endogenous estrogenic steroid), and potential endocrine disruptors (*).

Drinking water equivalent levels (DWELs) in μg/L were calculated using Equation (1), based on the ADIs (μg/kg day) developed in Reference [24]. Those ADIs were derived from the unobserved adverse effect level (NOAEL), or the lowest observed adverse effect level (LOAEL). A composite of uncertainty factors was applied to reflect uncertainties in extrapolation from experimental animals to humans, including the likely variation within the exposed population to account for significant gaps in the database used.

$$\text{DWEL} = \frac{\text{ADI} \times \text{BW}}{\text{IR}} \tag{1}$$

The ADIs derived in Reference [24] were combined with assumptions for the potential exposure for four age groups characterized by mean values for the drinking water ingestion rate (IR, L/day) and body weight (BW, kg) obtained from studies of the rural Mexican population [25] (Table 3).

**Table 3.** Exposure assumptions for rural Mexican population.

| Parameters | 2–4 Years Old | 5–11 Years Old | 12–15 Years Old | 20–70 Years Old | Reference |
|---|---|---|---|---|---|
| Ingestion Rate (L/d) | 0.938 | 1.43 | 1.79 | 1.84 | [25] |
| Body Weight (kg) | 14.13 | 30.56 | 53.66 | 67.8 | [25] |

The DWEL estimates represent the concentration in groundwater at/or below which adverse effects are not expected for human health. The age-dependent risk quotient (RQ) was then calculated for each of the EDCs by dividing the maximum measured concentration in groundwater (Cwg, μg/L) by the corresponding age-dependent drinking water equivalent level (DWEL) [Equation (2)].

$$\text{RQ} = \frac{C_{wg}}{\text{DWEL}} \tag{2}$$

For the characterization of risk, it was assumed that RQ > 1 indicates the possibility of risk for human health. If the RQ value was between 0.2 and 1, it called for more detailed assessment, whereas, if RQ ≤ 0.2, it was not considered an appreciable concern for human health [26].

## 3. Results and Discussion

### 3.1. Estimation of OMPs

A series of 17 micropollutants was described in the area for the different matrices (residual water and sources of supply) with at least nine compounds identified as endocrine disruptors in Reference [19] (Table 4), including 4-n-nonylphenol, di-2(ethilhexyl)-phthalate (DEHP), naproxen, and salicylic acid. These have been quantified by various studies on the wastewater originating in Mexico City [16,17],

which is also used for irrigation in the Tula Valley [19], conveyed via the "Central Emitter." The same works suggest this is the main route of entry to the environment of these compounds. The sewerage system represents the most common transport route for wastewater without pre-treatment generated in homes, hospitals, and industry.

**Table 4.** Limits of detection for organic compounds in residual water, supply sources, and instrument (IN). All in (ng/L).

| Phase | | LOQ | LOD | | Reference |
|---|---|---|---|---|---|
| | OMP | | RW | SS | |
| Acids | 2,4-D | 0.010 | 100,000 | 0.500 | [19] |
| | Clofibric acid | 0.010 | 100,000 | 0.500 | [19] |
| | Diclofenac | 0.010 | 50,000 | 1000 | [19] |
| | Gemfibrozil | 0.010 | 50,000 | 0.500 | [19] |
| | Ibuprofen | 0.010 | 50,000 | 0.250 | [19] |
| | Ketoprofen | 0.010 | 50,000 | 0.250 | [19] |
| | Naproxen | 0.005 | 50,000 | 0.250 | [19] |
| | Salicylic acid | 0.005 | 5000 | 0.250 | [19] |
| Phenols | 17α-ethinylestradiol * | 0.003 | 2500 | 0.050 | [19] |
| | 17β-estradiol * | 0.001 | 0.500 | 0.005 | [19] |
| | 4-n-nonylphenol * | 0.025 | 50,000 | 1000 | [19] |
| | Bisphenol-A * | 0.010 | 20,000 | 0.500 | [19] |
| | Butylbenzil-phthalate (BuBeP) * | 0.010 | 50,000 | 0.500 | [19] |
| | Di-2(ethilhexyl)-phthalate (DEHP) * | 0.010 | 50,000 | 0.005 | [19] |
| | Estrone * | 0.001 | 1000 | 0.005 | [19] |
| | Pentachlorophenol (PCP) * | 0.010 | 20,000 | 0.200 | [19] |
| | Triclosan * | 0.010 | 10,000 | 0.100 | [19] |

Limit of detection (LOD). Limit of quantitation (LOQ). Organic micropollutants (OMP). Residual water (RW). Supply sources (SS). Potential endocrine disruptors (*).

For this, matrix [19] reported a total of 17 organic compounds, and References [16,17] identified 18 compounds, which added carbamazepine to the previous list. A total of nine substances reported for wastewater and sources of supply have proven negative health implications because of their effects as endocrine disruptors on some organisms (Table 4). The mean concentrations in wastewater were: di-2(ethilhexyl)-phthalate (DEHP) (98,967 ng/L), 4-n-nonylphenol (9754 ng/L), bisphenol-A (1586 ng/L), butylbenzil-phthalate (BuBeP) (1331 ng/L), 2,4-Dichlorophenoxyacetic acid (2,4-D) (877 ng/L), and triclosan (748 ng/L). Other compounds occurred in concentrations of less than 150 ng/L: carbamazepine (144 ng/L), pentachlorophenol (69 ng/L), estrone (27 ng/L), 17β-estradiol (9 ng/L), and 17α-ethinylestradiol (3 ng/L). These results generally showed higher concentrations than those recorded in the literature (Figure 3).

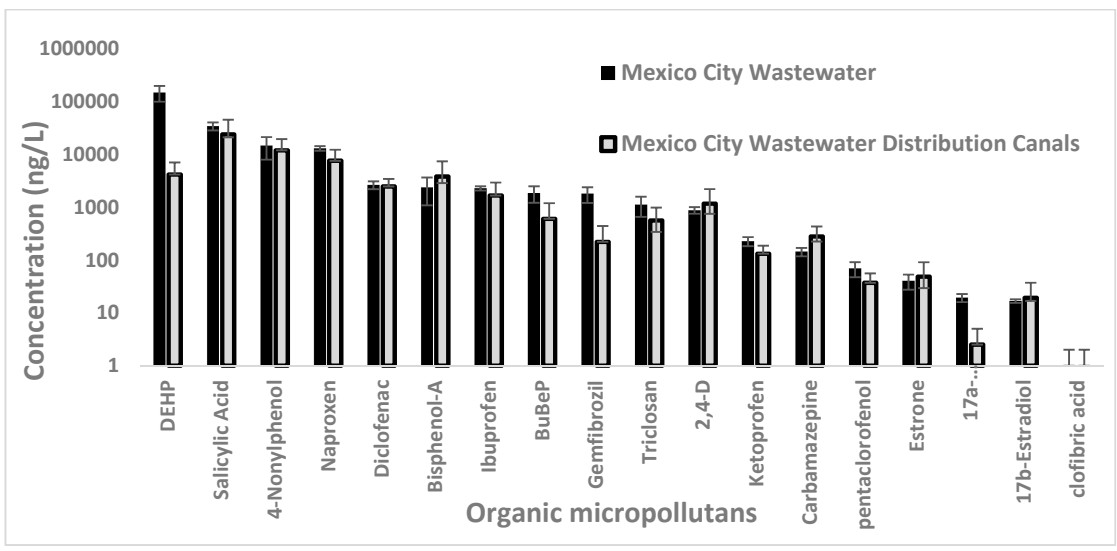

**Figure 3.** Presence of OMPs on wastewater in the distribution canals.

Once these substances arrive to the valley, they are spread throughout the primary canals (676.7 km) and the secondary canals (564.8 km). Eventually, they are used to supply the crops as dictated by their irrigation requirements. Most organic pollutants present few changes in their concentrations within the canals, as shown in Figure 3. However, in some cases, they increase or decrease in concentration significantly (as can be seen for gemfibrozil, DEHP, and 17α-ethinylestradiol). This phenomenon is attributed to the effects of water evaporation, dilution in the system, and other physical and chemical processes.

The quality of the wastewater changes during its transportation across the surface and by infiltration in the irrigation zone where the pollutants are transformed and transferred from one matrix to another (water to soil, or others). Most pollutants are retained in soil and in water through a complex interaction of natural attenuation mechanisms involving biodegradation, dilution, hydrolysis, infiltration, sorption, photolysis, precipitation, and infiltration. The previously mentioned processes occur naturally in water bodies, aquifers, soil, subsoil, and sediments. However, these processes depend on environmental, physicochemical, and biological conditions and properties, such as temperature, pressure, irradiation intensity, and some other conditions particular to each matrix [27]. During the transfer of wastewater to agricultural soil, water purification also occurs due to absorption (in plants and soil), oxidation, precipitation, and biological degradation.

In general, the concentration of pollutants in sources of the water supply was very low compared to the values measured in wastewater [28]. Studies carried out by References [28] and [29,30] in the Mexico Valley on compounds including 4-n-nonylphenol, carbamazepine, DEHP, ibuprofen, and naproxen showed that adsorption was the main mechanism of attenuation, due to the high content of organic matter and clays present in the valley. Triclosan was most rapidly removed from the wastewater (almost immediately). On the other hand, carbamazepine has been shown to be the compound most recalcitrant to biodegradation in soils and waters [29,31]. This compound has demonstrated the lowest sorption capacity when it occurs in natural sediments and also has the potential to reach groundwater under recharge conditions in semi-arid climates [32]. The following order of removal has been demonstrated, as shown in Table 5 (DEHF > triclosan > 4-n-nonylphenol > naproxen > estone > 17β-estradiol > ibuprofen > carbamazepine).

**Table 5.** Main attenuation mechanisms of analyzed OMPs.

| Pharmaceuticals | Log $k_w$ | $pk_a$ | Attenuation Mechanism in Water and/or Soil | Reference |
|---|---|---|---|---|
| 17α-ethinylestradiol * | 3.67/4.15 | 10.46 | Biodegradation/Adsorption | [33–37] |
| 17β-estradiol * | 4.01 | 10.71 | Biodegradation/adsorption | [34,36] |
| 2,4-Dichlorophenoxyacetic acid (2,4-D) | 2.81 | 2.64 | Biodegradation | [37] |
| 4-n-nonylphenol * | 4.48 | ND | Aerobic biodegradation | [37,38] |
| Bisphenol-A * | 3.4 | ND | Biodegradation/Sedimentation | [36] |
| Butylbenzil-phthalate * | 4.71 | ND | Aerobic Biodegradation/adsorption | [38] |
| Clofibric acid | 2.57 | 3 | Photolysis | [27] |
| Di-2(ethilhexyl)-phthalate (DEHP) * | 4.8–7.9 | ND | Adsorption | [36,39] |
| Diclofenac | 2.25 | 4.15 | Photolysis | [40] |
| Estrone * | 3.13 | 10.71 | Biodegradation/Adsorption | [36] |
| Gemfibrozil | 4.77 | ND | Photolysis/adsorption | [27] |
| Ibuprofen | 4-3.97 | 4.4–5.2 | Possible adsorption/sedimentation | [40] |
| Ketoprofen | 3.12 | 4.45 | Photolysis/adsorption | [27] |
| Naproxen | 3.18 | 4.15 | Photolysis | [27] |
| Pentachlorophenol * | 5.01 | 4.7 | Adsorption/photolysis (pH = 7.3) | [30,41] |
| Salicylic acid | 2.26 | 4.19 | Photolysis | [40] |
| Triclosan * | 4.8 | ND | Photolysis | [42] |

Kw (Dissociation constant of water), ND (Not determined), pKa (Acid dissociation constant), and potential endocrine disruptors (*).

As References [28,43] emphasize, the low soil sorption capacity and the high persistence of the compounds in the environment explain their high detection frequencies in groundwater of the Mezquital Valley. Although some compounds have been detected in trace concentrations in the aquifer, it could be argued that, for now, these concentrations do not represent a human health risk. However, care must be taken as the constant introduction of these compounds to the environment could, over time, surpass the natural treatment capability of the Tula Valley soil.

The principles described above are used as water treatment techniques known as "soil aquifer treatment" (SAT). This is applied in a controlled manner, by observing the same phenomena described for the Tula Valley. The occurrence of coupled processes during the treatment of wastewater appears to begin in Mexico City. However, it occurs in an uncontrolled manner. Initial concentration is an important factor to be considered in determining final concentrations in water that has undergone physicochemical processes, as suggested in Reference [4] for both purification phenomena (underground and surface transport) occurring in the Tula Valley. Similarly, studies in the area and reported in Reference [42] have established that the concentrations of the pharmaceuticals bezafibrate, clarithromycin, clindamycin, diclofenac, erythromycin, gemfibrozil, ibuprofen, metoprolol, naproxen, sulfasalazine, and trimethoprim are reduced along the course of wastewater transportation (canals and irrigation). This shows that anionic species (acidic pharmaceuticals) are capable of crossing the clay soils of the Tula Valley, so their removal is low in comparison to basic or neutral compounds (as confirmed by the present study).

This previously mentioned phenomenon occurs due to the transverse irrigation occurring in the region. The soil is likely to possess highly specialized mechanisms of biodegradation (derived from local microbiota and macrobiota) of endocrine disruptors present in irrigation water. These compounds may be rapidly degraded in the surface horizons of the soil, and, therefore, less likely to migrate to the aquifers. These natural attenuation processes are as effective as those achieved in wastewater treatment plants using sand filters under the same conditions, as suggested by Reference [15]. Soil removal rates of up to 90% were demonstrated for the OMPs evaluated. This is confirmed with the results shown in Figure 4 for the different sources of water located in the distribution canals. These results demonstrate removal rates of between 91.63% and 99.99% compared to those reported in Figure 3 (point of distribution).

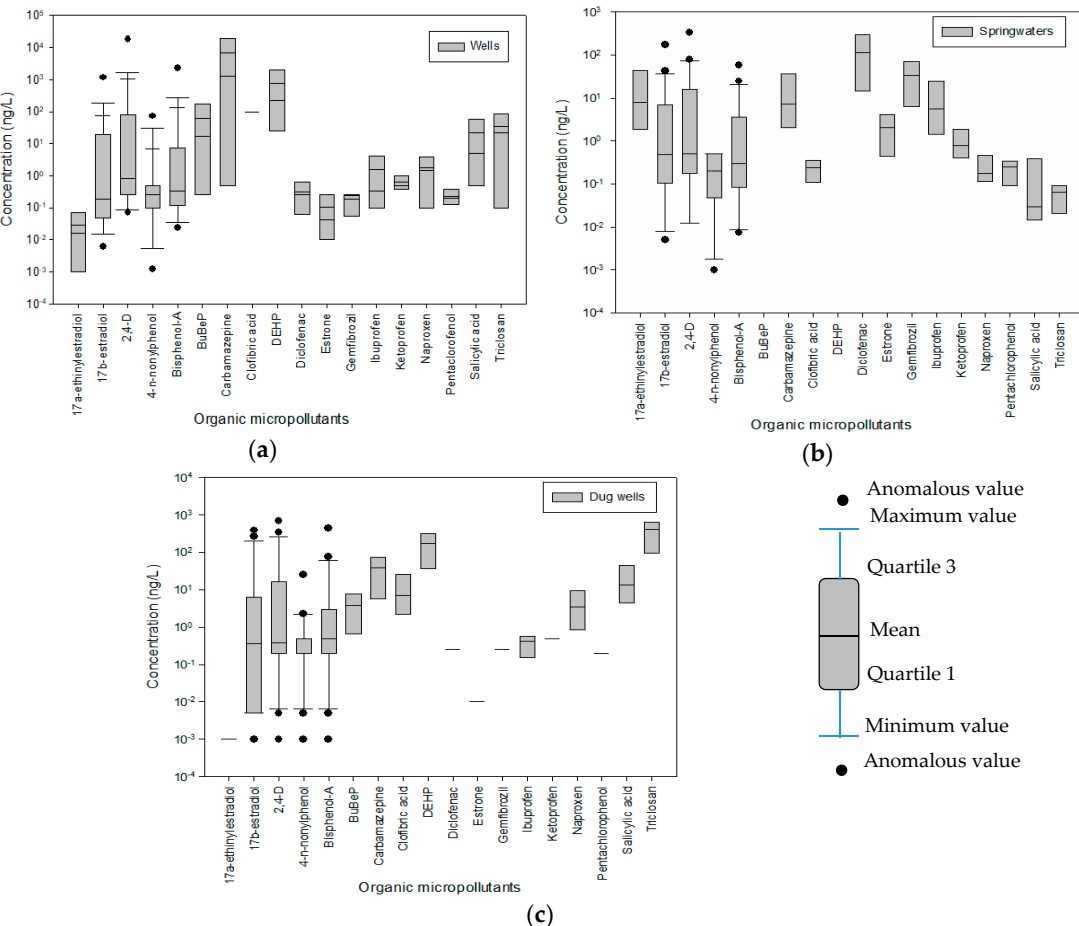

**Figure 4.** Presence of OMPs in Tula Valley in: (**a**) wells, (**b**) spring water, and (**c**) dug wells.

The differences in concentration for 2,4-D and carbamazepine recorded in wastewater and groundwater varies by a factor of between 19 and 19,508 times, respectively. However, their presence in the aquifer even in trace amounts, such as carbamazepine (4–12 ng/L), BuBeP (42–106 ng/L), bisphenol-A (6–17 ng/L), and triclosan (1–22 ng/L) may lead to adverse effects on exposed organisms with possible impacts on the transgenerational barrier.

The data presented for the distribution canals, wells, and spring water located within the Tula Valley show the importance of controlling and monitoring this water source, since the aquifer is used to supply 500 habitants of the adjacent locality due to the emergence of springs. It is imperative to establish the effect of pollutants on health and, if necessary, generate protection plans for the local communities and the environment itself. The removal of micropollutants in water prior to distribution or recharge (either incidental or planned) should be considered. Frequent monitoring of the quality for the aquifer should be carried out. Emphasis should be placed on the presence of endocrine disruptor compounds.

### 3.2. Estimation on Human Health Risks

The ADIs developed by Reference [24], designed to protect potentially exposed populations, were used in this study to estimate the predicted no-effect concentrations. The DWEL estimations for the maximum concentrations found in groundwater were used to measure the potential human risk in the Mezquital Valley. As summarized in Table 6, the ADIs used varied between 0.013 and 0.05 µg/kg-day (estrone and 17β-estradiol) to 100 and 570 µg/kg-day (BuBeP and naproxen). The DWEL estimates for children under 15 years of age are lower than adult values for all of the compounds assessed.

**Table 6.** Drinking water equivalent levels (DWELs) for Mezquital Valley groundwater.

| Phase | Compound | ADI (Reported) | 2–4 Years Old | 5–11 Years Old | 12–15 Years Old | 20–70 Years Old |
|---|---|---|---|---|---|---|
| | | µg/kg day | µg/kg day | µg/L | µg/L | µg/L |
| Acids | Diclofenac | 67 | 1009 | 1436 | 2006 | 2469 |
| | Gemfibrozil | 31 | 467 | 664 | 928 | 1142 |
| | Naproxen | 570 | 8588 | 12,215 | 17,062 | 21,003 |
| Phenols | 4-n-nonylphenol | 50 | 753 | 1071 | 1497 | 1842 |
| | 17β-estradiol | 0.05 | 0.75 | 1.07 | 1.50 | 1.84 |
| | Bisphenol-A | 50 | 753 | 1071 | 1497 | 1842 |
| | Butylbenzil-phthalate (BuBeP) | 100 | 1507 | 2,143 | 2,993 | 3,685 |
| | Carbamazepine | 10 | 151 | 214 | 299 | 368 |
| | Di-2(ethilhexyl)-phthalate (DEHP) | 12 | 181 | 257 | 359 | 442 |
| | Estrone | 0.013 | 0.20 | 0.28 | 0.39 | 0.48 |
| | Triclosan | 75 | 1130 | 1607 | 2245 | 2764 |

DWELs range from 0.20 µg/L (estrone for children < 4 years old) to 21,003 µg/L (naproxen for adults). In other words, estimates of DWELs indicate that, for the endogenous estrogenic steroid hormones, estrone, and 17β-estradiol, concentrations in groundwater of between 0.20 and 1.84 µg/L, are at or below values at which adverse human health effects are expected. However, for the other groups of EDCs, there exists an increase, greater than one order of magnitude, in the corresponding age-dependent drinking water equivalent levels (DWELs). Moreover, for the pharmaceutical compounds (carbamazepine, diclofenac, gemfibrozil, naproxen, and triclosan), acceptable concentrations vary between 151 µg/L and 21 mg/L. For industrial chemicals (4-n-nonylphenol, bisphenol-A, DEHP, and BuBeP), DWELs range from 181 µg/L to 4 mg/L.

DWELs were compared with the maximum groundwater concentrations (Cgw) found in the Mezquital Valley (Table 2). In order to show the potential risks, risk quotients were calculated (Table 3). These are all less than 1, with very low ratios of between $2.1 \times 10^{-7}$ and $1.3 \times 10^{-2}$. Therefore, there are no appreciable risks to human health as a result of the presence of the 11 compounds studied (Table 7), even for trace concentrations present in the source of drinking water in the Mezquital Valley.

**Table 7.** Age-dependent risk quotient (RQ) for Mezquital Valley groundwater.

| Phase | Compound | ADI (Reported) | 2–4 Years Old | 5–11 Years Old | 12–15 Years Old |
|---|---|---|---|---|---|
| | | µg/kg day | µg/kg day | µg/L | µg/L |
| Acids | Diclofenac | $6.14 \times 10^{-7}$ | $4.32 \times 10^{-7}$ | $3.09 \times 10^{-7}$ | $2.51 \times 10^{-7}$ |
| | Gemfibrozil | $5.14 \times 10^{-7}$ | $3.61 \times 10^{-7}$ | $2.59 \times 10^{-7}$ | $2.10 \times 10^{-7}$ |
| | Naproxen | $1.34 \times 10^{-6}$ | $9.41 \times 10^{-7}$ | $6.74 \times 10^{-7}$ | $5.48 \times 10^{-7}$ |
| Phenols | 4-n-nonylphenol | $1.00 \times 10^{-4}$ | $7.03 \times 10^{-5}$ | $5.04 \times 10^{-5}$ | $4.09 \times 10^{-5}$ |
| | 17β-estradiol | $9.29 \times 10^{-5}$ | $6.53 \times 10^{-5}$ | $4.68 \times 10^{-5}$ | $3.80 \times 10^{-5}$ |
| | Bisphenol-A | $2.27 \times 10^{-4}$ | $1.60 \times 10^{-4}$ | $1.14 \times 10^{-4}$ | $9.28 \times 10^{-5}$ |
| | Butylbenzil-phthalate (BuBeP) | $1.30 \times 10^{-2}$ | $9.12 \times 10^{-3}$ | $6.53 \times 10^{-3}$ | $5.31 \times 10^{-3}$ |
| | Carbamazepine | $1.16 \times 10^{-3}$ | $8.15 \times 10^{-4}$ | $5.84 \times 10^{-4}$ | $4.74 \times 10^{-4}$ |
| | Di-2(ethilhexyl)-phthalate (DEHP) | $1.12 \times 10^{-2}$ | $7.86 \times 10^{-3}$ | $5.63 \times 10^{-3}$ | $4.57 \times 10^{-3}$ |
| | Estrone | $1.33 \times 10^{-3}$ | $9.33 \times 10^{-4}$ | $6.68 \times 10^{-4}$ | $5.43 \times 10^{-4}$ |
| | Triclosan | $7.35 \times 10^{-5}$ | $5.16 \times 10^{-5}$ | $3.70 \times 10^{-5}$ | $3.00 \times 10^{-5}$ |

A recent review of EDCs in crops shows that the most studied are antibiotics [44,45]. Some studies present evidence of the toxic effects of EE2 and E2 in species of aquatic fauna, as well as the tendency for adsorption of these compounds onto soil or sediments. Their bioaccumulation and transport in the food chain (crops) requires further investigation [46]. It has been reported that there is no evidence of bisphenol-A passing into crops [47], since it was observed to remain in water or soil. The effects

of nonylphenols appear to be insignificant [12,48]. Phthalates have been extensively investigated in different agricultural areas of the world. They have been observed in soil and crops, and, in many cases, in concentrations within the acceptable limits of risk for human consumption [49]. Lastly, carbamazepine, which is among the extensive list of drugs reported in the literature, is possibly the most frequently studied due to its characteristics of persistence in the environment. Other studies have investigated ibuprofen, naproxen, and gemfibrozil, which are analyzed in the present research [50–52].

Their continued presence in water bodies, and the lack of water treatment infrastructure in the region could be a factor that would lead to an increase in concentrations of these compounds in sources of supply over time.

## 4. Conclusions

The raw wastewater that is sent from Mexico City (through the "Emisor Central") is used for agricultural irrigation in Tula Valley, and presents traces (ng/L) of at least 18 OMPs. Overall, in spite of the occurrence of some hard to degrade compounds in the groundwater of the Mezquital Valley aquifers, soil layers with a high content of organic matter and clays seem to act as filter and buffer systems for almost all of these compounds, which prevent contamination of the aquifer [18,28,43]. In addition, carbamazepine was regularly recorded, which suggests the possible presence of other polar and persistent organic micro-pollutants. Therefore, in order to take advantage of this water (potentially a very significant volume of water), a suitable treatment strategy including a process such as membrane filtration would be necessary to meet the quality requirements for potable supply to avoid future issues for subsequent water use. In this way, protection of the underlying aquifer will be achieved.

However, the retention capacity and the subsequent impact of the OMPs on water quality are still not completely understood. If the current use of water in irrigation is maintained (non-treated wastewater), future health issues could arise. Today, emerging micropollutants are still poorly studied and are generally unmonitored. Adverse effects on aquatic life and human health have been reported for some, but, for others, their effects remain unknown. This makes it necessary to, at least, study their characterization, location, distribution, and environmental interaction, as well as possible removal options using wastewater treatment systems.

The presence of trace concentrations of compounds such as bisphenol-A, carbamazepine, and triclosan in supply sources (shallow and deep aquifers) is worrying. It suggests the presence of other compounds that may or may not represent a potential chronic or immediate risk to the local population. However, there are no other studies being carried out presently in the zone for other OMPs or the possible impacts associated with water use.

The current risk assessment indicates that there is a substantial margin of safety for the consumption of low levels of EDCs in the water sources of the Mezquital Valley. It is very unlikely that these pose a risk for human health. The lack of studies in the area indicate the need for developing better approaches for evaluating the evidence, together with improved methods for risk assessment [14]. Traditional risk assessments may not always be appropriate when considering unresolved issues, including low-dose or non-threshold effects [53], or in determining how non-monotonic dose responses influence the way risk assessments are performed for chemicals with endocrine disrupting activities [54].

Investigations into the presence of EDCs in crops has largely been carried out in controlled experiments with treated wastewater. They tend to be studies conducted with high concentrations on laboratory or field scale, under controlled conditions. Usually, wastewater has undergone at least secondary treatment prior to its use for irrigation, other than, for several studies, which investigated the presence of phthalates. These scenarios are different from those that actually occur in the Mezquital Valley but are undoubtedly important references to consider in future research, which mainly concern factors that influence the transportation and observed destination of EDCs and analysis in soil, roots, leaves, and edible crops [55]. Yet, there are undoubtedly important references to consider in future research studies, which are mainly regarding the factors influencing the transport and observed

destination of EDCs [56,57], in an analytical experience for detection in soil, roots, leaves, and part of the edible cultivation [52], as well as in the risk assessment associated with the presence of EDCs in crops in the region [49,58].

Future research assessing the associated long-term risks and possible combined effects of chemical mixtures [14] would also be beneficial for an accurate exposure assessment to determine if there are any potential risks for human health.

Collectively, it will be possible to take appropriate measures to protect the human population from these harmful chemicals, as well as to facilitate better regulatory decision-making [21].

**Author Contributions:** The specific contributions of each author were: Conceptualization, A.C.C.-M. and B.E.J.-C. Methodology, A.C.C.-M., I.N.-G., and R.M.-L. Data validation, I.N.-G. and R.M.-L. Formal analysis, R.M.-L. Investigation, A.C.C.-M. and B.E.J.-C. Resources, A.C.C.-M. and B.E.J.-C. Data curation, D.U.-R. and P.I.Z.-S. Writing—A.C.C.-M., I.N.-G., D.U.-R., and R.M.-L. Writing—review and editing, R.M.-L. and P.I.Z.-S. Project administration, B.E.J.-C. Funding acquisition, A.C.C.-M. and B.E.J.-C.

**Funding:** The RECLAIM WATER Project under contract number 018309 in the Global Change and Eco-system sub-priority of the 6th Framework Programme funded this research.

**Acknowledgments:** We would like to acknowledge the European Commission (RECLAIM WATER, project number: 018309) for supporting this research.

**Conflicts of Interest:** The authors declare no conflict of interest.

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
