# Peer review of "Presence and Natural Treatment of Organic Micropollutants and their Risks after 100 Years of Incidental Water Reuse in Agricultural Irrigation"

_water, doi:10.3390/w11102148_

Round 1

Reviewer 1 Report

The presented manuscript is very interesting and important from environment protection point of view. Some items requires to be introduced and improved:

1.       There should be presented the analytical characteristics of used analytical procedure. The validation parameters such as linearity range, LOD, LOQ, repeatability and others.

2.       Table 2 and 3 – the additional column with references numbers is required.

3.       The detailed description of sampling points should be given: location, their distance from effluent of residual waters.

4.       Lines 155-164: the form of presentation of obtained results should be changed. Instead of their description, the results should be placed in a table. The table should include not only mean of determination but also the lowest and the highest results and median, frequencies of exact analyte determination. Were observed some differences in studied compounds concentrations in rain and dry seasons?

5.       The bars on Fig 3 should be completed with standard deviation.

6.       The information which are included in Table 4 are for nothing if they are not connected with real results obtained by Authors.

7.       Fig 4. – some explanation should be added: what is meaning of dots in Fig 4? Are presented results for one sampling point or are they mean for all sampling points from one studied area or for all studied areas? 

Author Response

Reply to Referees:

“The authors” express our respect to the referees for their valuable comments. After consider, and rework them, we have extended, explained and corrected all the observations over the manuscript, as each referee mentioned in a particular way. To continue the process, we would like to resubmit our manuscript once again for your consideration with all corrections effectuated.

Point 1. “Reviewer 1” comments: There should be presented the analytical characteristics of the used analytical procedure. The validation parameters such as linearity range, LOD, LOQ, repeatability, and others.

Response:

Regarding the comments about the validation parameters, Table 3 was modified specifying the validation parameters as requested. Regarding the analytical procedure is included its description between the lines 219-237 as was requested. Regarding the repeatability is included as was possible because the values were taken from the referred manuscripts. When the information is not contained in the original manuscript we cannot suppose it or derive it.

Point 2. “Reviewer 1” comments: Table 2 and 3 – the additional column with references numbers is required (check the manuscript please).

Response:

The additional column with the information for Table 2 and Table 3 was added as was requested.

Point 3. “Reviewer 1” comments: The detailed description of sampling points should be given: location, their distance from the effluent of residual waters.

Response:

A complete section of sampling was added and is located between the lines 178-218

Point 4. “Reviewer 1” comments: Lines 155-164: the form of presentation of obtained results should be changed. Instead of their description, the results should be placed in a table. The table should include not only mean of determination but also the lowest and the highest results and median, frequencies of exact analyte determination. Were observed some differences in studied compounds concentrations in the rain and dry seasons?

Response:

Figure 3 was modified to represent the results obtained considering statistics parameters as was suggested. We consider that emphasizing numerically as text on some micropollutants are very important because some of them are not appreciable in numbers. Regarding the seasonal analysis: The proposed data mining for the manuscript was made as a global study. A seasonal data exploration is considered but with other purposes, is included and deeply analyzed with some complementary data by another unpublished work.

Point 5. “Reviewer 1” comments: The bars on Fig 3 should be completed with standard deviation.

Response:

Figure 3 was modified to represent the results obtained considering the standard deviation for each micropollutant analyzed.

Point 6. “Reviewer 1” comments:  The information which is included in Table 4 are for nothing if they are not connected with real results obtained by Authors.

Response:

Table 4 was renamed as Table 6. The results are discussed between the lines 319-356.

Point 7. “Reviewer 1” comments: Fig 4. – some explanation should be added: what is the meaning of dots in Fig 4? Are presented results for one sampling point or are they mean for all sampling points from one studied area or for all studied areas?

Response:

 The explanation of the sampling points is now extended between the lines 178-218 and between the lines 357-385

Note: Professional translation services were used for manuscript

Reviewer 2 Report

The manuscript presents routine work on the monitoring of micropollutants in Mexico City Mezquital Valley and in agricultural soil and aquifers, the findings were not comprehensively discussed. This manuscript evaluated 18 micropollutants in water/soil, which is a general monitoring without any scientific experimental design. This paper in its present form does not bring new relevant information to the field. There should be some solid theoretical evidence provided to support your proposed idea which should lead to some scientifically significant results.

Introduction should be rewritten. It should be expanded to include a more detailed discussion of current problems, and potential applications.

Would you explicitly specify the novelty of your work? What progress against the most recent state-of-the-art similar studies was made?

The literature review section should be improved. It should be dedicated to present critical analysis of state-of-the-art related work to justify the objective of the study. Also, critical comments should be made on the results of the cited works.

The discussion statements are speculations. More detailed discussion of factors affecting the distribution of the micropollutants should be added. Make every attempt to improve the discussion by critically analyzing your findings.

Conclusions should be amended to incorporate a broader discussion of the significance and potential application of this specific study.

Author Response

Reviewer 2

Reply to Referees:

“The authors” express our respect to the referees for their valuable comments. After consider, and rework them, we have extended, explained and corrected all the observations over the manuscript, as each referee mentioned in a particular way. To continue the process, we would like to resubmit our manuscript once again for your consideration with all corrections effectuated.

Point 1. “Reviewer 2” comments: The manuscript presents routine work on the monitoring of micropollutants in Mexico City Mezquital Valley and in agricultural soil and aquifers, the findings were not comprehensively discussed. This manuscript evaluated 18 micropollutants in water/soil, which is general monitoring without any scientific experimental design. This paper in its present form does not bring new relevant information to the field. There should be some solid theoretical evidence provided to support your proposed idea which should lead to some scientifically significant results.

Response:

The relevance of the findings is now included in the introduction. Indeed, most of the works for the zone were developed by the same authors. There are not many references for the zone of study. Nowadays the Mezquital valley has become a very important zone in terms of water for legislation, economics, politics, environmental, treatment, reuses, etc.

Point 2. Introduction should be rewritten. It should be expanded to include a more detailed discussion of current problems, and potential applications.

Response:

The Introduction has been improved as required.

Point 3. Would you explicitly specify the novelty of your work? What progress against the most recent state-of-the-art similar studies was made?

Response:

The novelty of the work is spread all over the text. No previous work in Mexico has evaluated such a large area of influence (as an integral system), nevertheless in a specific way it is detailed between the lines 51-95, 110-120, 136-151, 152-175, Regarding the progress is related to the integral analysis of the system (85,000 ha), and the analysis of risk

Point 4. The literature review section should be improved. It should be dedicated to present a critical analysis of state-of-the-art related work to justify the objective of the study. Also, critical comments should be made on the results of the cited works.

Response:

Regarding the state of art, previous works are still scarce as we mentioned (lines 136-144). All the works for the zone are focused on making a checklist (lines 140-144). This work tries to assess the system and its impact for several purposes (lines 152-175)

Point 5. The discussion statements are speculations. A more detailed discussion of factors affecting the distribution of the micropollutants should be added. Make every attempt to improve the discussion by critically analyzing your findings.

This is the first work made for the area (in this way) and the information is scarce. We are only capable to make some inferences with the evidence of other cited manuscripts, moreover, none of them are opposed with what we raised (or with the findings of the other manuscripts) and with the integral analysis suggests the findings as we describe (inferential analysis). Nevertheless, it is true that it is necessary to carry out more studies in the area to corroborate or to rebut, and we are working on it for the following papers.

Point 6. Conclusions should be amended to incorporate a broader discussion of the significance and potential application of this specific study

The conclusions have been improved as required (lines 433-488)

General points:

All the work has been improved (abstract, introduction, methodology, results and discussion, and conclusions). In addition, professional translating services were used for the manuscript

Reviewer 3 Report

The authors have presented their work on the presence and natural treatment of emerging micropollutants in Mexico City's water supply due to reuse of wastewater for irrigation purposes. The authors have done a excellent job of tabulating the detected micropollutants in the Mezquital Valley in regards to the complex water basin (i.e. channels, tunnels, wells, springs, etc.) of the study area. The report also thoroughly presents the location-based micropollutant concentrations as well as the daily intake and relative risk due to these concentrations. While the technical content is compelling, there are several grammar issues (i.e., missing punctuation marks, awkward sentence structures) throughout the manuscript should be addressed. After another session of proofreading is made to make necessary grammar revisions, I believe this manuscript will be acceptable for publication.

A few example of grammar errors are as follows:

53. Awkward sentence structure and in passive voice.

56. Missing semicolon.

93. Missing comma and passive voice should be changed to active.

138-139. Sentence fragment and 'if' must be lowercase after a semicolon. 

168. Missing comma.

Author Response

Reviewer 3

Reply to Referees:

“The authors” express our respect to the referees for their valuable comments. After consider, and rework them, we have extended, explained and corrected all the observations over the manuscript, as each referee mentioned in a particular way. To continue the process, we would like to resubmit our manuscript once again for your consideration with all corrections effectuated.

Point 1. There are several grammar issues (i.e., missing punctuation marks, awkward sentence structures) throughout the manuscript should be addressed. After another session of proofreading is made to make necessary grammar revisions, I believe this manuscript will be acceptable for publication.

Response:

All the work has been improved (abstract, introduction, methodology, results and discussion, and conclusions). In addition, professional translation services were used for the manuscript

Reviewer 4 Report

The study is presenting a concrete health risk case study about potential OMP pollution from untreated RW used in irrigation on the source of water for drinking purposes. This study is definitely of interest and would deserved to be published after a more in-depth analysis, more accurate information on the sampling protocol and improved English style are performed.

Introduction:

Could you precise/confirm the current treatment applied to WW at the moment?? Is it really raw sewage without even primary/pretreatment? Different treatment depending on the WW origin? Is this WW directly applied for irrigation of there is a filtration/storage phase first?

Section 2:

how many samples??? How many times/sampling campaigns per season?  Please precise also you sampling process in this section. In fact there should be a section dedicated to sampling which is an important factor in your study.

Section 3:

Table 3: why limit of detection is much higher in RW?

L167-173: please make a more in-depth analysis. Could you make calculation to estimate the rate of dilution/concentration based on evaporation/dilution and son on and see if it fits with the hypothesis.

In the overall discussion, what is missing is a graph/table explaining the rate of elimination of all the OMPs. You should discuss what is the removal rate of all the elements since it is not very clear the way it is presented.

Also, evolution of concentration of OMPs is related to accumulation (with years) or just the consequence of what is coming from RWW? Could you discuss that aspect?

The health risk study is based on water consumption as far as I understand. However, if I understand well, the WW used is for irrigation and therefore there is a very high risk of OMP accumulation on the plants that are used de facto to treat the WW. Could you please integrate this in the discussion since it is a main risk issue and possibly refer to health risk studies dedicated to consumption of plants by humans/animals in the region (or in other case studies)

Few comments regarding the English and structure of the document. Please revise and review by a native speaker since sometimes the manuscript is hard to understand. Also check the referencing style. See below some examples of what should be improved:

L34: ''e''stablish L38: reference needed. L44-45: weird sentence! L54: ‘’to assess the strength of evidence’’ ??? What do you mean? L88: ´´conducted by’’ do not put just the reference, sentence incomplete…(several times through the document) add or the authors names xxxx et al. or something like ‘’conducted in several studies’’ Table 1 format: please adjust.

Author Response

Reviewer 4

Reply to Referees:

“The authors” express our respect to the referees for their valuable comments. After consider, and rework them, we have extended, explained and corrected all the observations over the manuscript, as each referee mentioned in a particular way. To continue the process, we would like to resubmit our manuscript once again for your consideration with all corrections effectuated.

Point 1. The study is presenting a concrete health risk case study about potential OMP pollution from untreated RW used in irrigation on the source of water for drinking purposes. This study is definitely of interest and would deserved to be published after a more in-depth analysis, more accurate information on the sampling protocol and improved English style are performed.

Response:

A complete section of sampling was added and is located between the lines 178-218 Professional translation services were used for the manuscript to improve the language used for the manuscript

Point 2. Introduction: Could you precise/confirm the current treatment applied to WW at the moment?? Is it really raw sewage without even primary/pretreatment? Different treatment depending on the WW origin? Is this WW directly applied for irrigation of there is a filtration/storage phase first?

Response:

A deeper explanation was made to answer the questions made by the “Reviewer 4”. The introduction and material and methods were improved to answer them. The reviewer can find them between the lines 37-175 and lines 177-237.

Point 3. Section 2: how many samples??? How many times/sampling campaigns per season?  Please precise also you sampling process in this section. In fact there should be a section dedicated to sampling which is an important factor in your study.

Response:

A complete section of sampling was added and is located between the lines 178-218

Point 4. Section 3: L167-173: please make a more in-depth analysis. Could you make calculation to estimate the rate of dilution/concentration based on evaporation/dilution and son on and see if it fits with the hypothesis.

Response:

It is not possible for this manuscript because of the total area of study is about 85,000 ha, and the studies for the Mezquital valley are still scarce as we emphasize.

Point 5. Section 3: In the overall discussion, what is missing is a graph/table explaining the rate of elimination of all the OMPs. You should discuss what is the removal rate of all the elements since it is not very clear the way it is presented.

Response:

Figure 3 represents the removal of each OMP, to improve a better understanding a deeper and extended explanation is included between the lines 285-356 as requested

Point 5. Section 3: Also, evolution of concentration of OMPs is related to accumulation (with years) or just the consequence of what is coming from RWW? Could you discuss that aspect?

The evolution of the concentration occurs as a consequence of a combination of the accumulation regulated by natural mechanisms of attenuation and the RWW, nevertheless, this occurs in a differential way depending on the zone because of the water-route (as we mentioned during the text)

Point 6. Section 3: The health risk study is based on water consumption as far as I understand. However, if I understand well, the WW used is for irrigation and therefore there is a very high risk of OMP accumulation on the plants that are used de facto to treat the WW. Could you please integrate this in the discussion since it is the main risk issue and possibly refer to health risk studies dedicated to the consumption of plants by humans/animals in the region (or in other case studies)

In general studies for the zone are scarce, nevertheless, the information available is between the lines 136-151, 162-167, 335-341, 415-431 and 446-459

Point 7. Section 3: L34: ''e''stablish L38: reference needed. L44-45: weird sentence! L54: ‘’to assess the strength of evidence’’ ??? What do you mean? L88: ´´conducted by’’ do not put just the reference, sentence incomplete…(several times through the document) add or the authors names xxxx et al. or something like ‘’conducted in several studies’’ Table 1 format: please adjust. 

The typos mentioned and others were improved by professional translation services.

Round 2

Reviewer 1 Report

In my opinion, the revised manuscript is fully acceptable for publication.

Reviewer 4 Report

The manuscript have been impressively improved with regards to all my main comments and is ready for publication now.